# Dissimilar Laser Welding of Austenitic Stainless Steel and Abrasion-Resistant Steel: Microstructural Evolution and Mechanical Properties Enhanced by Post-Weld Heat Treatment

**DOI:** 10.3390/ma14195580

**Published:** 2021-09-26

**Authors:** Mikko Hietala, Matias Jaskari, Mohammed Ali, Antti Järvenpää, Atef Hamada

**Affiliations:** 1Kerttu Saalasti Institute, University of Oulu, FI-85500 Nivala, Finland; mikko.hietala@oulu.fi (M.H.); matias.jaskari@oulu.fi (M.J.); antti.jarvenpaa@oulu.fi (A.J.); 2Materials and Mechanical Engineering Unit, University of Oulu, Pentti Kaiteran Katu 1, FI-90014 Oulu, Finland; mohammed.ali@oulu.fi; 3Steel Technology Department, Central Metallurgical Research and Development Institute, Helwan 11421, Egypt

**Keywords:** laser welding, abrasion resistant steel, austenitic stainless steel, butt joint, post-weld heat treatment, electron backscatter diffraction

## Abstract

In this study, ultra-high-strength steels, namely, cold-hardened austenitic stainless steel AISI 301 and martensitic abrasion-resistant steel AR600, as base metals (BMs) were butt-welded using a disk laser to evaluate the microstructure, mechanical properties, and effect of post-weld heat treatment (PWHT) at 250 °C of the dissimilar joints. The welding processes were conducted at different energy inputs (EIs; 50–320 J/mm). The microstructural evolution of the fusion zones (FZ) in the welded joints was examined using electron backscattering diffraction (EBSD) and laser scanning confocal microscopy. The hardness profiles across the weldments and tensile properties of the as-welded joints and the corresponding PWHT joints were measured using a microhardness tester and universal material testing equipment. The EBSD results showed that the microstructures of the welded joints were relatively similar since the microstructure of the FZ was composed of a lath martensite matrix with a small fraction of austenite. The welded structure exhibited significantly higher microhardness at the lower EIs of 50 and 100 J/mm (640 HV). However, tempered martensite was promoted at the high EI of 320 J/mm, significantly reducing the hardness of the FZ to 520 HV. The mechanical tensile properties were considerably affected by the EI of the as-welded joints. Moreover, the PWHT enhanced the tensile properties by increasing the deformation capacity due to promoting the tempered martensite in the FZ.

## 1. Introduction

Ultra-high-strength structural steels (UHSS) are widely used in different engineering applications, such as the automotive industry, engineering machinery, mine exploitation, military, and aerospace industry, because of their good performance. The use of UHSS in automotive manufacturing has the greatest potential to decrease car weight and increase the safety factor. In vehicles, weight reduction is directly utilized as a way to lower emissions due to lower fuel consumption. The mass reduction of structures is achieved by employing thin sheets of high-strength steel [1,2]. 

In modern lightweight structures, there is a need to efficiently join two different sheets of UHSS together. The joining of dissimilar steels has advantages due to their special material properties that can be utilized in the structures. Since the joining of dissimilar steels offers large challenges for the designers of the structures, more information is needed on the characteristics of the dissimilar joints welded using a laser. Laser welding of dissimilar steels is known as an excellent method of joining dissimilar steels with different physical properties [3,4].

Abrasion-resistant (AR) steels are classified as ultra-high-strength steels due to their superior strength under static loading and high hardness. The utilization of abrasion-resistant steel is growing, especially in the earthmoving and mining industries [5]. However, there is a lack of information on weldability, microstructural evolution, and the correlated properties of dissimilar joints between AR steel and stainless steel (SS). Recently, we studied the optimization of the tensile shear strength of laser-welded lap joints of ultra-high-strength AR steel [6]. The study showed the very fine microstructure characteristics of the weld, such as prior austenite grains and the fresh martensite. It was found that the energy input significantly affected the grain size of the fresh martensite.

The austenitic stainless steel (ASS) 300 series is the most common stainless steel family due to their superior corrosion resistance and high mechanical properties at both room and elevated temperatures. Hence, ASS is used in several industrial applications and different structures that require pairing with a dissimilar metal. Several reports presented the dissimilar welding of ASS with low-strength C-steel. For instance, Cam et al. [7] studied the tensile strength of dissimilar joints between ASS and structural C-steel. They found that the tensile strength of the joints fractured at the lower-strength base metal sides. They also observed that dissimilar joints showed an inhomogeneous weld metal microstructure that contained ferritic–austenitic and bainitic and martensitic phases.

The difference in thermal conductivities of austenitic stainless steel and carbon steel poses challenges in the laser welding of dissimilar butt joints since the differences in the cooling rates of steels induce residual stresses in the joint. Moreover, welding defects could be induced in the weldments due to the various heat absorption of the pairing steels [8]. It was reported that the heat absorptivity of the stainless steel of Nd:YAG laser radiation is more than three times higher than that of the carbon steel [9].

Moreover, the different physico-chemical properties of the paired metals in the dissimilar welded joints induce a significant structural inhomogeneity [10]. Pańcikiewicz et al. [11] reported that the difference in the melting temperature and thermal conductivity coefficients between dissimilar metals induces uneven melting during laser-induced dissimilar welding. Landowski et al. [12] showed that the difference in the thermal conductivity coefficient between two dissimilar stainless steel, e.g., 316L austenitic and 2304 lean duplex stainless steels, could promote a non-uniform phase structure in the weldments.

Prabakaran et al. [13] studied the effects of post-weld heat treatment (PWHT) on the mechanical properties, the microstructure, and the chemical composition of the laser weld. They found that the PWHT improved the tensile strength of joints only marginally. Sumi et al. [14] investigated the effects of chemical composition on the microstructure of laser-welded joints of ultra-high-strength steels. They found that the toughness of the weld was highly dependent on the carbon content of the welded steels. 

Gnanasekaran et al. [15] studied the effect of laser power on the tensile properties of laser-welded joints of austenitic stainless steel (AISI 301). They found that the higher the energy input, the higher the tensile strength of the joints. Anawa et al. [16] investigated the optimization of the tensile strength of laser-welded dissimilar joints of ASS (AISI 304) and carbon steel. They reported that the tensile strength of the joints was significantly affected by the welding parameters, such as the laser power and welding speed. Rossini et al. [17] investigated the dissimilar laser welding of advanced high-strength steel sheets (DP, 22MnB5, and TRIP steels) for the automotive industry. The microstructural analysis of the weld zone showed a full martensite structure for all the dissimilar joints.

This study aimed to demonstrate the industrial reliability of the laser welding of paired metals of UHSS sheets with a 3 mm thickness and various chemical compositions and different phase structures. In this work, dissimilar butt joints between cold-worked austenitic stainless steel with a strength of 1.3 GPa (AISI 301) and abrasion-resistant steel with a strength of 2.2 GPa (AR600) were manufactured using laser welding with various EIs (50-320 J/mm). PWHT at a low temperature of 250 °C was applied to enhance the microstructure of the weld zone. The mechanical properties of the tensile strength and distribution of the microhardness across the weld zone were conducted for the as-welded and PWHT joints.

## 2. Materials and Methods

Ultra-high-strength steels sheets with a 3 mm thickness, namely, cold-hardened austenitic stainless steel AISI 301 (ASS) and martensitic abrasion-resistant steel AR600, were welded using a laser to manufacture dissimilar butt joints. The ASS and AR600 steel sheets were delivered by Outokumpu Stainless Oy (Tornio, Finland) and SSAB Europe Oy (Raahe, Finland), respectively. The chemical compositions of the BMs, which were sourced by the suppliers, are shown in Table 1. The mechanical properties of the BMs, as measured using uniaxial tensile testing, are illustrated in Table 2. The AR600 exhibited a higher tensile strength of 2.2 GPa with a lower ductility of 6%. This was attributed to the hard martensitic matrix of the AR600. However, the ASS showed a lower strength of 1.3 GPa and higher ductility of 22%. This was attributed to the deformed structure of the austenitic matrix, which had coexisting deformed austenite and strain-induced α′-martensite structures. 

A Yb: YAG diode-pumped disk laser (model: Trumpf HDL-4002, Ditzingen, Germany) with the following specifications: continuous wavelength of 1030 nm, the beam quality of 8 mm∗mrad, and a maximum power of 4 kW was employed to manufacture the dissimilar butt joints between the ASS and AR600. Constant laser parameters, such as a laser defocus of −1 mm and a laser spot of 0.3 mm, were applied in the experiments. The typical dimensions of each steel plate used for the dissimilar joints are 300 mm × 150 mm × 3 mm.

The welding process was autogenous, i.e., without filler metal, and undergone with various energy inputs (EI) from extremely low EI of 50 J/mm to the highest EI of 320 J/mm. The welding parameters are illustrated in Table 3. The welding process was conducted under argon shielding gas (ArSG) with a gas flow rate of 30 L/min. The ArSG of purity 99.99% was blown through five nozzles in the welding direction to ensure shielding and protection of the FZ. The direction of welding was in the transverse to the rolling direction of the BMs. The PWHT of the laser-welded butt joints was carried out in a muffle oven (Nabertherm GmbH: Germany). The joints were placed in the preheated oven at 250 °C with isothermal holding for 1 h, followed by final air cooling to room temperature. It was established that, depending on the tempering temperature and time, the evolution of the martensitic structure may vary significantly during tempering since at tempering temperatures below 250 °C, interstitial atoms, such as C atoms, are mobile. However, the substitutional elements are almost immobile due to their very low diffusivities. Hence, tempering at 250 °C leads to precipitation of ε-iron carbide and the recovery of a dislocation structure in martensite [18,19].

In accordance with the ASTM E8/8M standard, tensile specimens with a gauge length of 75 mm and width of 12.5 mm were then machined perpendicular to the welding direction. A Zwick Z 100 tensile machine (Zwick Roell, GmbH: Germany) was used for tensile testing at a constant strain rate of 10^−3^ s^−1^. Three samples of each butt joint were tensile tested for each EI. A Micro Vickers Hardness Tester (Innovatest Falcon 511: Netherlands) was used to measure the microhardness profiles of the weldments using a load of 0.2 kg. 

The weld morphologies that were induced at the different EIs, such as weld size and geometry, were depicted using a laser scanning confocal microscope (model: KEYENCE/VK-X200: Osaka, Japan). The microstructural characteristics of the FZ and heat-affected zone (HAZ) were studied using a field-emission gun scanning electron microscope (FEG-SEM) (model: Carl Zeiss Ultra plus: Oberkochen, Germany) that was equipped with electron backscatter diffraction (EBSD) capability. For the metallographic examination using EBSD, the ASS/AR600 welds were sectioned perpendicular to the welding direction and the transverse cross-section was metallographically prepared according to the standard preparation technique, i.e., mechanically polished using SiC papers, then down to 1 μm by using a diamond suspension, and finally chemically polished using a 0.05 μm colloidal suspension of silica for about 10 min. For the laser scanning confocal microscopy examination, the welds were chemically etched using 2% nital.

We utilized MATLAB software, along with the MTEX texture and the crystallographic toolbox, to reconstruct the original austenite grains, i.e., prior austenite grain (PAG), which transformed into martensite during solidification of the melt pool of the weld zone, as described in [20,21].

## 3. Results and Discussion

### 3.1. Weld Morphology of Laser-Welded Butt Joints with Different Energy Inputs

The macroscopic morphology of the dissimilar welded joints with different EIs is shown in Figure 1. A gentle etching was applied to highlight and identify the different zones of the weldments, such as the FZ and HAZs in the paired metals. It was observed that different-colored zones were revealed on both sides of the FZ, which is highlighted by the white lines. As expected, the sizes of the FZ and the HAZ were significantly influenced by the EI since the sizes of the FZ and HAZ were gradually increased with the EI, as shown in Figure 1. At the EI 50 J/mm, the average size of the FZ was ~0.99 mm and the corresponding HAZs in the ASS and AR600 were 0.30 and 0.29, respectively, as shown in Figure 1a. Moreover, the lines of the weld seams were observed in the FZ. When the EI was increased to 100 J/mm (Figure 1b), the FZ size increased to 1.225 mm and the HAZs were wider in the paired metals with sizes of 0.385 and 0.425 mm. It is emphasized that one of the major concerns of high-quality welding in the industry is the manufacture of dissimilar joints without severe cracks. Since the AR600 steel used in the current experiments had a medium carbon content (0.33% C), it was a crack-sensitive metal for fusion welding. As a result, microcracks and porosity were found in some of the laser welds. For instance, micropores were observed in the FZ achieved at an EI of 100 J/mm, as shown in Figure 1b.

When the laser welding was conducted at the higher EIs of 160 and 320 J/mm, the FZs and HAZs were significantly wider, as shown in Figure 1c,d. The size of the HAZ at the EI of 320 J/mm is shown in the upper-right corner in Figure 1d. The dependence of the FZ width on the EI indicated that there was an increase in the molten pool when the EI was increased. In agreement, Chen et al. [22] reported a similar dependence of the weld width on the heat input of laser-welded 800 MPa grade high-strength low-alloy steel.

### 3.2. Microstructure Analysis

Figure 2a shows the image quality (IQ) map of the FZ at an EI of 50 J/mm. It can be observed that two regions with different grain structures of the BM-ASS on the left side are visible. The HAZ was composed of fine equiaxed grains that underwent recrystallization that was caused by the heat transfer from the weld pool. However, the grain structure of the BM-ASS was deformed coarse grains. Similarly, the grain structure of HAZ in the BM-AR600 was relatively affected by increasing the grain size, as marked by the yellow line. Furthermore, the IQ map of the FZ microstructure, highlighted by the white lines in Figure 2a, clearly revealed elongated grains with a dark grayscale level. The corresponding phase map showed two different domains with different colors, as shown in Figure 2b. The red domain was an α′-*bct* structure and the blue domain was a γ-*fcc* structure. It was observed that two phases coexisted, namely, γ-*fcc* + α′-*bct,* in the FZ in the middle. 

At a high magnification of the FZ, the EBSD misorientation boundary mapping was undertaken (Figure 3a). A high density of low-angle grain boundaries LAGBs (green) was displayed in the martensite grains. This was in agreement with Morito et al. [23]. They reported that misorientation angles across boundaries within the martensite packets and the boundaries between the sub-blocks are LAGBs. The corresponding phase map showed the predominant phase in the FZ was α′-*bct* martensite (red), as shown in Figure 3b.

The relative frequencies of LAGBs (green; 2–15°) and high-angle boundaries (dark; >15°) seen in the FZ microstructure (Figure 3a) are plotted in Figure 3c. It was observed that the relative frequency of LAGBs was significantly high in the FZ. This suggested that the martensite structure was the dominant phase in the weld metal of the butt joints between the ASS and AR600.

Figure 4 illustrates the microstructures of the HAZ and the fusion boundary (FB) region at the side of the BM-ASS. As shown in the IQ map (Figure 4a), the HAZ exhibited new recrystallized austenitic grains. The FB showed elongated grains toward the BM. Interestingly, these grains at the FB were not fully transformed to α′-*bct* martensite, as shown in the phase map in Figure 4b. This was attributed to the temperature gradients and the solidification rate in the FB region adjacent to the BM.

The general view of the weld metal at EI 160 J/mm is shown in Figure 5. The IQ map displays that the greyscale level of the FZ was similar to that of the BM-AR600. However, the HAZ of the BM-ASS side revealed visible fine and coarse austenitic grains, as shown in Figure 5a. Moreover, the size of the FZ at that EI was wider than the counterpart at the lower EI of 50 J/mm in Figure 2. It is well known that the melt area and the size of the melt pool increase with the EI during welding. The corresponding phase map (Figure 5b) shows a duplex structure of α′-*bct* and γ-*fcc* austenite in the FZ.

Similarly, Figure 6 reveals the microstructures of the HAZ and the FB region at the side of the BM-ASS at an EI of 160 J/mm. As shown in the IQ map (Figure 6a), the FB, highlighted by yellow lines, showed coarse elongated austenitic grains along with α′-*bct* martensite, which is similar to the lath martensite in the FZ. The HAZ exhibited new equiaxed recrystallized austenitic grains. It was observed that an α′-*bct* martensite with different morphology coexisted inside the new austenitic grains, as shown in the phase map in Figure 6b. It is well reported that the welding process promotes residual stresses caused by high temperature gradients and fast cooling rates. Consequently, the accumulation of the residual stresses achieved by laser welding could increase its tendency toward a solid-state phase transformation due to the low stacking fault energy of the fcc structure of the studied BM-ASS [24,25]. In other words, stress-induced martensite formation (SIMF) was promoted in the HAZ of the BM-ASS due to the welding residual stress.

As the EI increases to 320 J/mm, the IQ map of the wide FZ, shown in Figure 7a, clearly revealed a typical lath martensite structure. However, the corresponding phase map showed that the austenite phase coexisted and was linearly distributed within the fresh martensitic matrix, as shown in Figure 7b. This weld structure promoted by dissimilar welding of the ASS and AR600 was similar to the martensitic structure of the abrasion-resistant steels bearing small fractions of retained austenite, which transformed into martensite during mechanical deformation. Consequently, the work hardening capability of the steel increased due to the TRIP effect [26].

The microstructures of the HAZ and the FB region at the side of the BM-ASS at an EI of 320 J/mm are shown in Figure 8. The FB, highlighted by yellow lines, showed coarse elongated austenitic grains, along with an α′-*bct* martensite. Interestingly, the HAZ exhibited coarse recrystallized austenitic grains without inducing α′-*bct* martensite via a solid-state transformation as in the lower EI. It is reasonable to assume that the residual stress that occurred due to the welding process at the high EI did not reach the critical value to promote the stress-induced martensite formation.

The measured phase structure of the weld metal that was produced through the dissimilar welding between the studied steels, AISI 301 and AR600, was compared with the predicted phase structure using a Schaeffler diagram by assuming the equal contribution of the paired plates steels in the weld metal, i.e., 50% dilution of AR600. By calculating the chromium and nickel equivalents of the steels and plotting these values on the Schaeffler diagram, the predicted structure of the weld metal could be determined. It was observed that the predicted weld structure contained a small amount of austenite and predominantly martensite, as shown in Figure 9. Hence, the measured phase structure agreed well with the predicted structure.

It can be observed from Figure 10 that the size and morphology of the PAG were significantly affected by the EI. It was observed that the morphology of the PAGs was mainly columnar, with the average sizes of 40 ± 3, 61 ± 4, 76 ± 6, and 100 ± 9 μm at EIs of 50, 100, 160, and 320 J/mm, respectively. Interestingly, the PAGs were divided into two regions with different grain morphologies. This was clearly visible at the lower EIs of 50 and 100 J/mm (Figure 10a,b). One region displayed small, elongated grains and the other contained large, elongated grains. This was attributed to the different thermo-physical properties of the dissimilar paired metals and inevitably in the chemistry of each metal. The thermal conductivity of the stainless steel is significantly higher than that of the carbon steel, namely, AR600, at high temperatures >1000 °C [28]. Accordingly, the cooling rate of the ASS side is higher than that of the AR600. Hence, it is reasonable to assume that the zone of small elongated PAGs is promoted adjacent to the BM-ASS.

### 3.3. Hardness

The hardness distribution profiles across the dissimilar welds of the paired metals, namely, ASS and AR600, are shown in Figure 11. The corresponding hardness measurements of the PWHT samples were inserted to show the influence of that treatment on the hardness. It was apparent that the energy input dependence of the hardness distribution showed various trends. Two striking characteristics were shown in the hardness profiles of the as-welded samples, as shown with red lines. First, there was a significant variation in the hardness distribution across the FZ. This was attributed to the coexistence of the soft γ-*fcc* in the hard austenitic matrix, as shown in the EBSD maps (Figure 3, Figure 5 and Figure 7). Second, at the lower EIs of 50 and 100 J/mm, it was observed that the average hardness of the as-welded FZ was approximately comparable with that of the BM-AR600. The hardness of the AR600 BM was nearly 645 HV because of the fully martensitic structure. However, when the EI was increased to 160 and 320 J/mm, the hardness of the FZ decreased to 600 and 520 HV, respectively. Several researchers reported the softening of the FZ during high EIs due to promoting tempered martensite with a low dislocation density [6,29,30].

The microhardness of the softened zones, namely, the HAZs, on both sides of the FZ was considerably lower compared to the BMs. For instance, the hardness of the ASS BM was 415 HV. At an EI of 320 J/mm, the hardness of the HAZ of the ASS was 220 HV, i.e., about a 50% reduction in the hardness of the softened zone. This was attributed to activating the softening mechanism that promoted new soft austenite grains, i.e., recrystallization.

When applying the PWHT at 250 °C, the fresh martensitic of the FZ underwent a low-temperature tempering treatment since, during the PWHT, the martensite tended to undergo extensive recovery, i.e., dislocation rearrangement and annihilation was activated within this structure. Consequently, recovered martensite with a low dislocation density was induced along with carbide precipitates. The tempered martensitic structure (TM) was associated with a reduction in hardness, as shown from the hardness profiles (black lines in Figure 11). At the high EI of 320 J/mm (Figure 11d), the hardness profiles of the as-welded and PWHT samples were perfectly matched in the FZ and HAZ. This means that a similar structure of TM was promoted in the as-welded and PWHT samples.

### 3.4. Tensile Strength

The dissimilar butt joints were characterized using conventional quasi-static tensile straining, as well as PWHT joints. The mechanical properties of the BMs are illustrated in Table 2. The quasi-static flow curves of the dissimilar joints are shown in Figure 12. It was observed that the shape of the flow curve of the as-welded dissimilar joint depended on the EI, as shown in Figure 12a. For instance, the shape of the tensile curve of the dissimilar joint welded with the lowest EI of 50 J/mm was a complete stress–strain curve displaying elastic–plastic deformation behavior with total elongation of 17%. This shows that the fracture took place with a necking phenomenon and was located at the ASS BM. At an EI of 100 J/mm, the plastic deformation capacity was significantly reduced with total elongation of 5%. With the higher EIs of 160 and 320 J/mm, the plastic deformation capacity was insignificant since the fracture position was located in the HAZ of the AR600 BM. Recently, we studied the tensile strength and fracture surface of the dissimilar ASS/AR600 joints that were welded with two EIs: 160 and 320 J/mm [31]. The dissimilar joints failed at the location of the FZ away from the BMs. Clear evidence of the brittle fracture with relatively smooth surfaces was shown on the fracture surfaces. In other words, cleavage fracture across the lath martensite boundaries was the predominant feature of the fracture surface.

By applying the PWHT at 250 °C, the mechanical tensile properties of the dissimilar joints were enhanced, as shown in Figure 12b. The shape of the flow curves of the joints welded with different EIs was relatively similar to that of the ASS BM since, the fracture position was located at the ASS BM, as shown in Figure 13. However, at an EI of 320 J/mm, the softening of the HAZ in the ASS was significantly higher with a low hardness of 220 HV (see Figure 11d). Hence, the fracture was located at that softening zone of the ASS, as shown in Figure 13. Consequently, the PWHT joint at that EI displayed a lower yield strength of 780 MPa.

## 4. Conclusions

In this study, ultra-high-strength steels, i.e., 1.3 GPa strength cold-hardened austenitic stainless steel AISI 301 (ASS) and 2.2 GPa strength martensitic abrasion-resistant steel AR600, were welded using laser welding with different energy inputs (EIs) (50–320 J/mm). The microstructure and mechanical properties of the dissimilar butt joints were studied; the main conclusions are as follows:

(1) Sound dissimilar welded joints of 301ASS/AR600 laser welding were manufactured with no severe welding defects, such as pores and microcracks, formed in the fusion zone. The microstructure of the FZ was composed of lath martensite and a small fraction of austenite.

(2) The average microhardness of the welded joint decreased with increasing the energy input from 640 HV at an EI of 50 J/mm to 520 HV at an EI of 320 J/mm. Furthermore, the microhardness in the fusion zone was relatively nonuniform due to the coexistence of the soft phase γ-*fcc* within the hard martensitic α′-*bct* matrix.

(3) The mechanical tensile properties, strength, and elongation of the as-welded dissimilar joints decreased with an increase in the EI from 50 to 320 J/mm. When the EI was 50–100 J/mm, fracturing occurred in the ASS BM, while with the higher EI of 160–320 J/mm, fracturing occurred in the HAZ of the AR600 BM.

(4) The PWHT at 250 °C for 1 h enhanced the mechanical properties of the dissimilar joints at the various studied EIs by increasing the deformation capacity due to promoting tempered martensite structure in the FZ. Consequently, the strength and elongation were similar to that of the 301 ASS BM.

## Figures and Tables

**Figure 1 materials-14-05580-f001:**
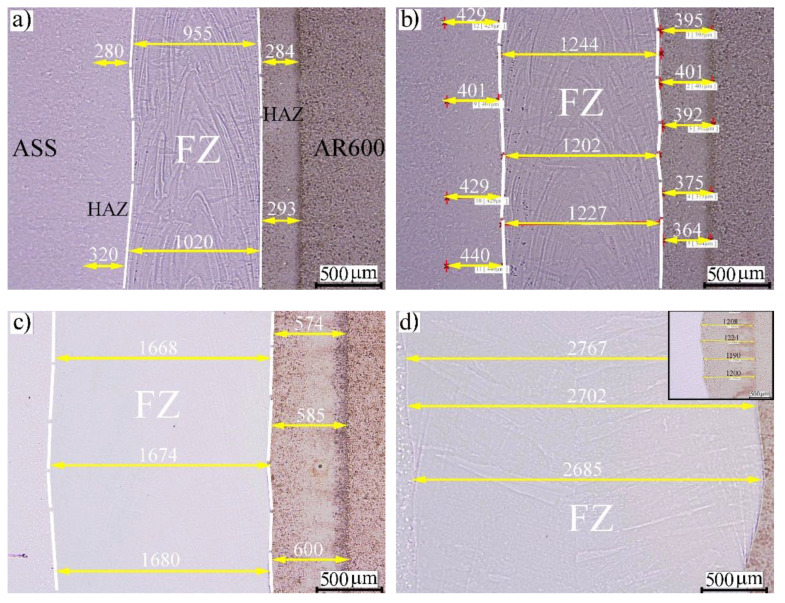
Morphology of welded joints with various EIs: (**a**) 50 J/mm; (**b**) 100 J/mm; (**c**) 160 J/mm; (**d**) 320 J/mm.

**Figure 2 materials-14-05580-f002:**
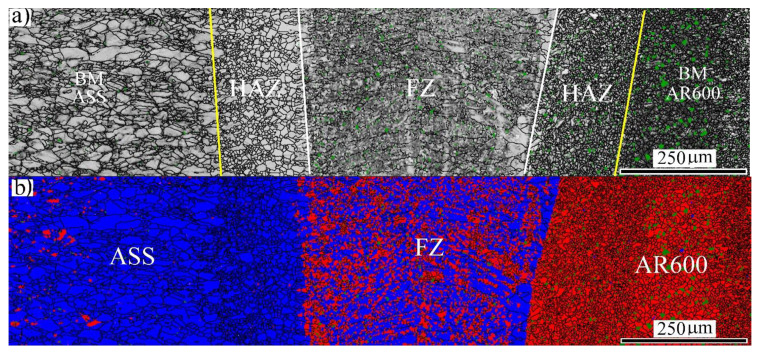
General view of the microstructure across the weld section of the dissimilar laser-welded butt joint at 50 J/mm: (**a**) EBSD-IQ map; (**b**) EBSD phase map with indexing α′-*bct* in red and γ-*fcc* austenite in blue.

**Figure 3 materials-14-05580-f003:**
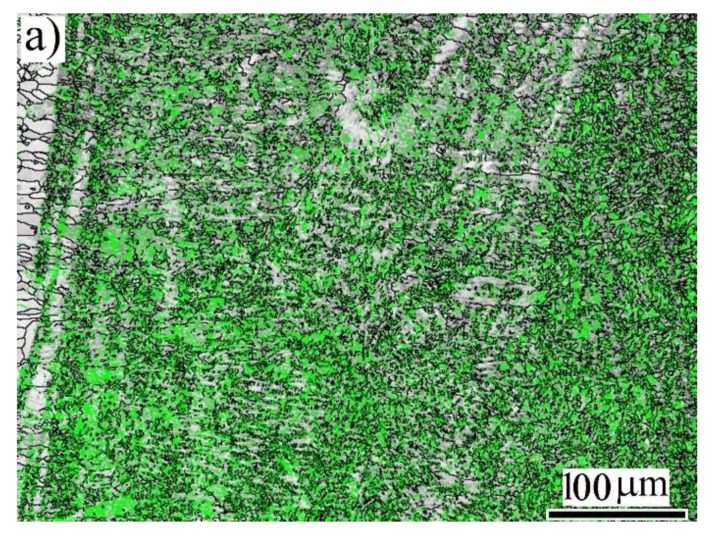
Close view of the FZ structure at EI 50 J/mm: (**a**) EBSD misorientation map (low-angle boundaries in green); (**b**) EBSD phase map with indexing α′-*bct* in red and γ-*fcc* austenite in blue; (**c**) relative frequency of low- and high-angle grain boundaries corresponding to (**a**).

**Figure 4 materials-14-05580-f004:**
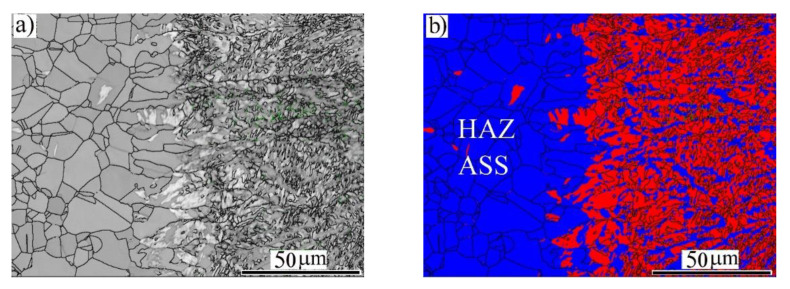
EBSD imaging maps of the fusion zone (FZ) near the fusion boundary (FB) of the BM-ASS at EI 50 J/mm: (**a**) image quality (IQ) map; (**b**) phase map α′-*bct* in red and γ-*fcc* austenite in blue.

**Figure 5 materials-14-05580-f005:**
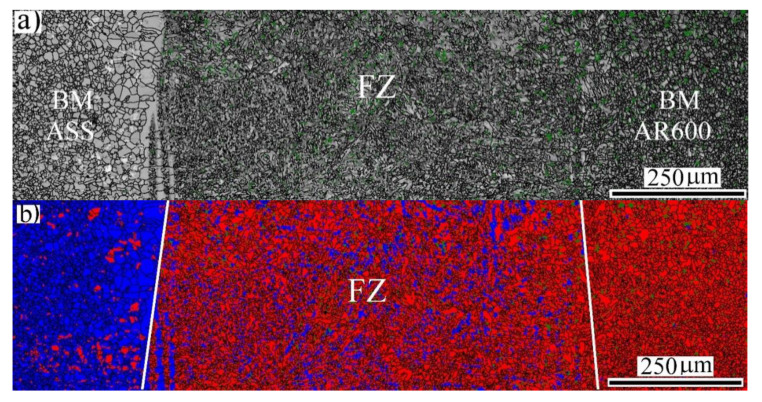
General view of the microstructure across the weld section of the dissimilar laser-welded butt joint at an EI of 160 J/mm: (**a**) EBSD IQ map; (**b**) EBSD phase map with indexing α′-*bct* in red and γ-*fcc* austenite in blue.

**Figure 6 materials-14-05580-f006:**
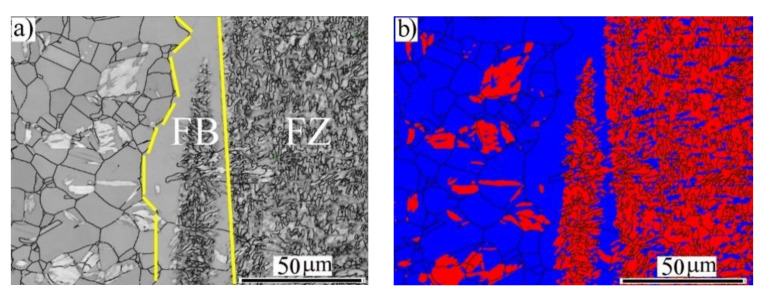
EBSD imaging maps of the fusion zone (FZ) near the fusion boundary (FB) of the BM-ASS at an EI of 160 J/mm: (**a**) image quality (IQ) map; (**b**) phase map of α′-*bct* in red and γ-*fcc* austenite in blue.

**Figure 7 materials-14-05580-f007:**
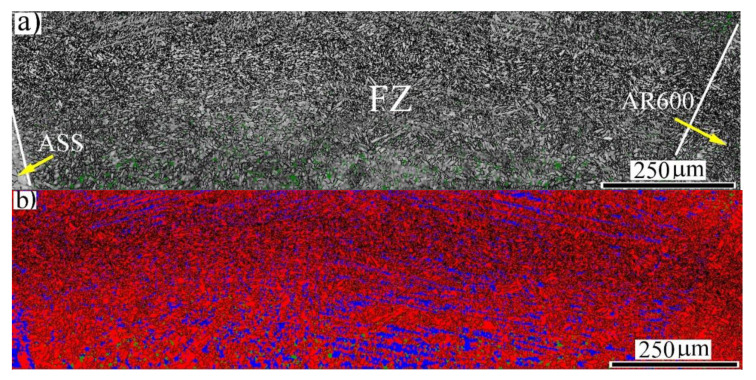
General view of the microstructure across the weld section of the dissimilar laser-welded butt joint at 320 J/mm: (**a**) EBSD IQ map; (**b**) EBSD phase map with indexing of α′-*bct* in red and γ-*fcc* austenite in blue.

**Figure 8 materials-14-05580-f008:**
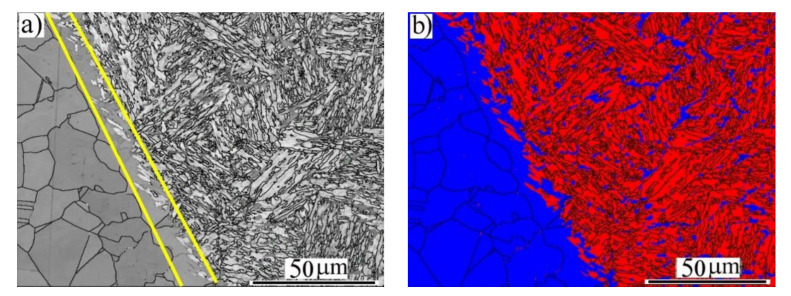
EBSD imaging maps of the fusion zone (FZ) near the fusion boundary (FB) of the BM-ASS at an EI of 320 J/mm: (**a**) image quality (IQ) map; (**b**) phase map of α′-*bct* in red and γ-*fcc* austenite in blue.

**Figure 9 materials-14-05580-f009:**
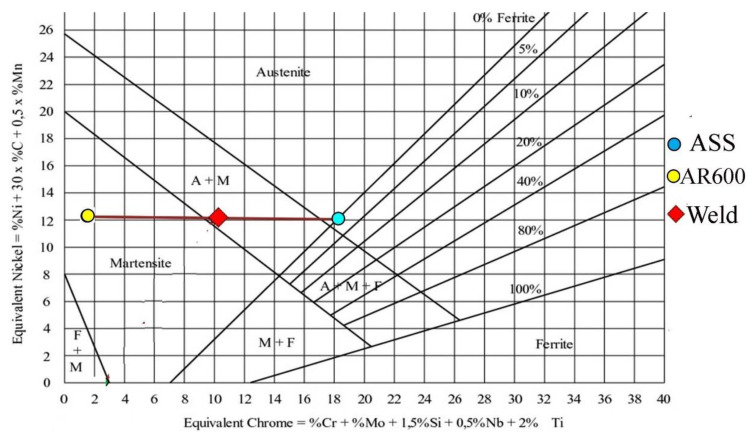
A Schaeffler diagram of the predicted structure of the studied dissimilar weld metal between austenitic stainless steel 301 and martensitic C-steel AR600 that is described in the current manuscript. The diagram was taken from [27].

**Figure 10 materials-14-05580-f010:**
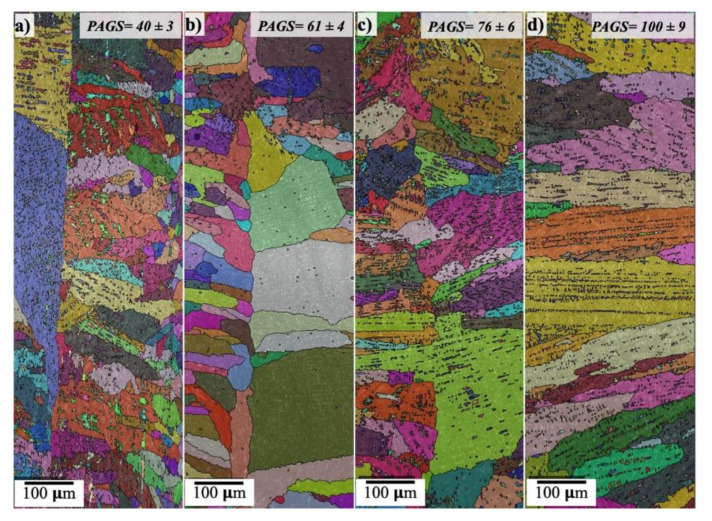
EBSD orientation maps superimposed with IQ maps for the prior austenite grains of the weld metal in the fusion zone at different laser EIs: (**a**) 50 J/mm; (**b**) 100 J/mm; (**c**) 160 J/mm; (**d**) 320 J/mm. The reconstructed PAG maps were created using MTEX and MATLAB software.

**Figure 11 materials-14-05580-f011:**
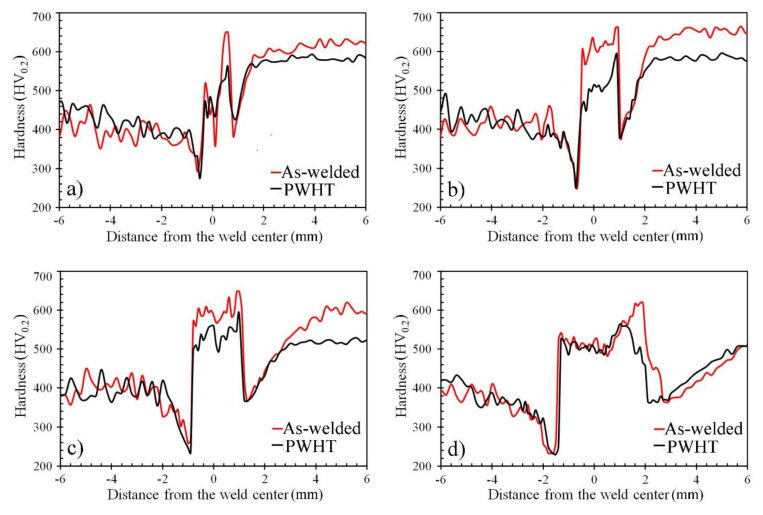
The hardness profiles of the dissimilar laser-welded butt joints between the ASS and AR600 at different EIs without and with post-welding heat treatment (PWHT) at 250 °C for 1 h: (**a**) 50 J/mm; (**b**) 100 J/mm; (**c**) 160 J/mm; (**d**) 320 J/mm.

**Figure 12 materials-14-05580-f012:**
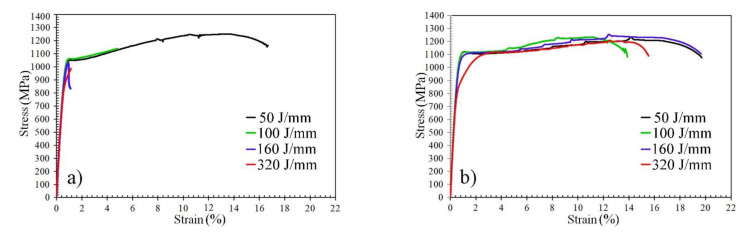
Typical engineering strain–stress curves of the dissimilar butt joints ASS/AR600 with different EIs: (**a**) as-welded; (**b**) with post-weld heat treatment at 250 °C for 1 h.

**Figure 13 materials-14-05580-f013:**
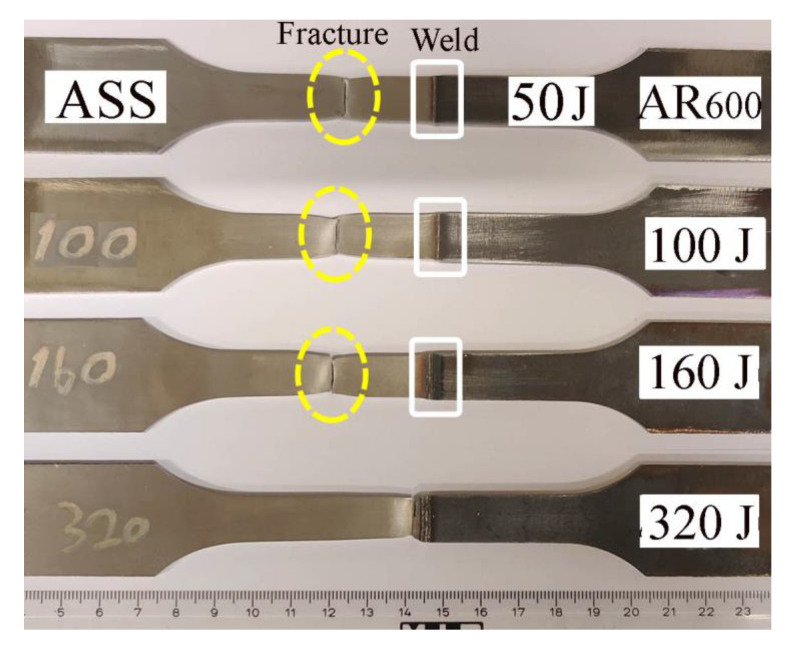
Macrograph images of the tensile samples of the dissimilar joints after testing showing the fraction positions in the post-weld joints that were heat-treated at 250 °C.

**Table 1 materials-14-05580-t001:** Chemical composition (%) of the base metals.

Steel Code	C	Si	Mn	P	S	Cr	Ni	Mo	N	Cu
ASS	0.106	0.85	1.21	0.030	0.001	16.76	6.51	0.32	0.060	0.340
AR600	0.326	0.31	0.49	0.009	0.001	0.81	2.01	0.48	0.003	0.014

**Table 2 materials-14-05580-t002:** Mechanical properties of the experimental base metals under quasi-static tensile tests.

Steel Code	YS (MPa)	UTS (MPa)	Elongation (%)	HV_0.2_
ASS	914 ± 19	1307 ± 17	21.5 ± 0.7	416 ± 14
AR600	1774 ± 24	2164 ± 14	6.4 ± 0.2	646 ± 9

**Table 3 materials-14-05580-t003:** Laser welding parameters.

Laser Power (kW)	Welding Speed (m/min)	Energy Input (J/mm)
4	4.80	50
4	2.40	100
4	1.50	160
4	0.75	320

## Data Availability

All the data generated during this study are included in this article.

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
