# Peer review of "Dissimilar Laser Welding of Austenitic Stainless Steel and Abrasion-Resistant Steel: Microstructural Evolution and Mechanical Properties Enhanced by Post-Weld Heat Treatment"

_materials, 2021, doi:10.3390/ma14195580_

Round 1

Reviewer 1 Report

In my opinion, the approach of this topic concerning the ultra-high-strength structural steels (UHSS), the effect of post-weld heat treatment (PWHT) on the microstructure and mechanical properties of dissimilar joints, is interesting in terms of developing a defect detection method; this work contains experimental data achieved by various techniques and reported relative to other investigations; the achieved results are extensively discussed.

The linguistic quality of the text is very good, the expression is clear, concise and without mistakes, so the scientific message is conveyed with clarity.

As a result, I would recommend some minor revisions of this manuscript due to the following reasons:

  1. Page 10 Paragraphs 312-314 “Accordingly, the cooling rate of ASS side is hinger than that of the AR600. It is resealable to assume that the zone of small elongated PAGs is promoted adjacent to the BM-ASS.”

Please correct the expression errors.

  1. Page 11 Paragraphs 340-343 “With applying PWHT at 250 °C, the fresh martensitic of the FZ is undergone low-temperature tempering, in which decomposition of martensite in which recovery of martensite lath with low dislocation density and the precipitation of carbides are promoted”

Please rephrase the sentence.

  1. Page 11 Section 3.4. Paragraph “The dissimilar butt joints were characterized by conventional quasi-static tensile straining as well as PWHT joints. Mechanical properties are illustrated in Table 3.”

The correct number of Table displaying mechanical properties it is 2.

Final Conclusion: The paper meets the necessary standards for publication provided that the above said minor corrections are implemented.

Reviewer 2 Report

Dear Authors,

The reviewed work titled: “Dissimilar laser welding of an austenitic stainless steel and abrasion resistant steel: Microstructural evolution and mechanical properties enhanced by post-weld heat treatment” describes an experimental study aimed at assessing the weldability (morphology and properties) of dissimilar laser steel welded joints (before and after heat treatment). Below I present the comments that should be introduced into the manuscript before deciding whether to accept the manuscript for publication.

  1. From the point of view of the readability of the work, it is advantageous to consistently describe and show the joints. Please arrange the descriptions in the title and content of the manuscript: always the same material on the left (i.e. the first described). Now, for example, AR600 is: second in the title, first in Fig. 11 and second in Fig. 13.
  2. Abstract: I propose to swap (switch places) sentences in lines: 20-23. It is now reported that fracture morphology observations were first performed and then tensile tests.
  3. The Introduction is well prepared. Please consider supplementing paragraph 67-72 with another weldability limitations of dissimilar joints with the use of the current literature: https://doi.org/10.3390/met10050559, https://doi.org/10.3390/ma13204540 https://doi.org/10.3390/ma13132930
  4. Chapter 2:

What were the dimensions of the samples? Please add that the welding process was "autogenous". What are sources of the data listed in tables 1 and 2?

How is quantity "energy input" defined in the manuscript? Is it equivalent to "heat input"?

Which Ar was used (designation, purity)? Was it fed from both sides of the joints?

Please describe all devices in accordance with the journal’s guidelines.

There is no description of heat treatment in this chapter: device-furnace, heating and cooling rate. On what basis were the temperature and heat treatment time selected?

Have VT tests been carried out? Is it possible to show photos of the joints?

How the samples for metallographic tests were prepared?

  1. Results

Figure 3: the font is too small.

Figure 7: Please mark the steel symbols on the photos.

Lines 300-303: this information should be in advance, in chapter 2.

Lines 328: please verify correctness: “Figs. (2-8)"

Figure 12 a: Legend font is too small.

  1. Authors Contributions: who did the research works?
  2. Conflict of Interests: 408-434 - please delete any unnecessary text.

Round 2

Reviewer 2 Report

Dear Authors, 

Thank you very much for the accurate and professional replies and explanations and for introducing changes in line with my comments. I have no further comments. I believe that your article will be a very good source of information for readers.

Author Response

There are no further comments from Reviewer 2. He recommended the work for publication. He said that:

"Thank you very much for the accurate and professional replies and explanations and for introducing changes in line with my comments. I have no further comments. I believe that your article will be a very good source of information for readers"